# Room-temperature pyro-catalytic hydrogen generation of 2D few-layer black phosphorene under cold-hot alternation

Huilin You[1], Yanmin Jia[1], Zheng Wu[2], Feifei Wang[3], Haitao Huang [4] & Yu Wang[5]

Many 2D few-layer materials show piezoelectric or pyroelectric effects due to the loss-of-inversion symmetry induced by broken structure, although they are not piezoelectric or pyroelectric in the bulk. In this work, we find that the puckered graphene-like 2D few-layer black phosphorene is pyroelectric and shows a pyro-catalytic effect, where the pyroelectric charges generated under ambient cold–hot alternation are utilized for hydrogen evolution and dye molecule decomposition. Under thermal cycling between 15 °C and 65 °C, the 2D few-layer black phosphorene shows a direct hydrogen generation of about 540 µmol per gram of catalyst after 24 thermal cycles and about 99% decomposition of Rhodamine B dye after 5 thermal cycles. This work opens a door for the pyro-catalytic energy harvesting from the cold–hot alternations by a class of 2D few-layer materials.

[1] Department of Physics, Zhejiang Normal University, Jinhua 321004, China. [2] College of Geography and Environmental Sciences, Zhejiang Normal University, Jinhua 321004, China. [3] Key Laboratory of Optoelectronic Material and Device, Department of Physics, Shanghai Normal University, Shanghai 200235, China. [4] Department of Applied Physics, The Hong Kong Polytechnic University, Hong Kong, China. [5] School of Materials Science and Engineering, Nanchang University, Nanchang 330031, China. Correspondence and requests for materials should be addressed to Y.J. (email: ymjia@zjnu.edu.cn) or to Z.W. (email: wuzheng@zjnu.edu.cn) or to H.H. (email: aphhuang@polyu.edu.hk)

Two-dimension (2D) materials have become one of the most interesting research areas owing to the emergence of 2D graphene with a honeycombed structure in recent years[1–3]. A new mono-elementary 2D material, named 2D few-layer black phosphorene (2D-BP), has been reported as a magic material showing unexpected anisotropic optoelectronic and electronic properties, and has attracted global interest[3–5]. For example, a high carrier mobility of the 2D-BP has been predicted in theory and measured successfully[6,7]. It has also been reported that the monolayer black phosphorene is very stable and maintains an ordered puckered hexagonal structure up to a <300 °C temperature[8]. The band gap of few-layer 2D-BP changes from 2.0 to 0.7 eV with increasing number of layers[9,10]. When more layers are added, the band gap further reduces and eventually reaches ~0.3 eV for bulk black phosphorus due to the interaction among different layers[3,11]. The charge confinement within the normal direction of the thin layers is regarded as the reason for this thickness dependent band gap[6].

The black phosphorene exhibits almost the same intra-group and the inter-group bond lengths of around 2.22 Å between P atoms[12]. The inter-group and intra-group bond angles are 102.09° and 96.36°, respectively[13]. All of these geometric parameters are used to establish the 2D-BP structure as shown in Fig.1.

Generally, bulk black phosphorus processes a centro-symmetric crystal structure with a point group of $mmm$[14]. As shown in Fig. 1, the centro-symmetry is broken in 2D-BP, which shows a non-centro-symmetric structure with an $m$ point group. Therefore, piezoelectric or pyroelectric effects are expected in the 2D-BP with a broken inversion symmetry[15,16]. In general, ferroelectric materials are a sub-group of pyroelectrics and all pyroelectrics are piezoelectric[17]. Many few-layer materials ($MoSe_2$, $MoS_2$, and others) show strong piezoelectric behavior, whereas their bulk crystals are non-ferroelectric or non-piezoelectric due to centro-symmetry[2,15,16]. Hu et al. predicted an in-plane ferroelectric polarization in atomic-thick phosphorene nanoribbons by using the first-principles calculation[18]. Ong et al. reported a strong piezoelectric effect in 2D few-layer graphene with adatoms[19], whose point group symmetry changes from 6/$mmm$ to $mm$2 (orthorhombic structure) or $m$ (monoclinic structure), as calculated by density functional theory and observed experimentally[19].

The detailed components $p_1$, $p_2$, and $p_3$ of the pyroelectric coefficient vector $p$ ($p = \{p_i\}$, $i = 1, 2, 3$ denotes the orthogonal base vectors) can be simply deduced from Neumann's Principle, which states that the physical properties and the corresponding crystal structure will follow the same symmetry operation. For $m$ point group, we have,

$$p_j = \alpha_{ji} p_i, \tag{1}$$

where the $p_j$ is the pyroelectric coefficient vector after symmetry operation. The footnote $j$ denotes the numerically subscripted symbols 1, 2, and 3 of the orthonormal base vectors. The transformational matrix $\alpha_{ji}$ for the $m$ point group can be expressed as Eq. (2),

$$\alpha_{ji} = \begin{pmatrix} 1 & 0 & 0 \\ 0 & -1 & 0 \\ 0 & 0 & 1 \end{pmatrix}. \tag{2}$$

Combing Eqs. (1) and (2), we obtain the pyroelectric coefficients $p_1 \neq 0$, $p_2 = 0$, and $p_3 \neq 0$.

Experimentally, it is difficult to directly measure the pyroelectric charge response of a single-black phosphorene nanosheet under a temperature alternation excitation, due to the limit of nanosize. Similar to the case of photocatalysis where photo-generated charges participate the electrochemical redox reactions, it is intuitive to expect a pyro-catalytic effect in 2D-BP, where the pyroelectric charges can take part in the redox reactions under cold–hot temperature alternation excitation. Xie et al. reported the coupling between pyroelectric effect and electrochemical process and observed its water-splitting effect driven by these pyroelectric charges[20]. Belitz et al. investigated the $BaTiO_3$ microcrystalline powders for pyro-driven hydrogen generation[21]. Kakekhani et al. theoretically studied the pyroelectrically induced water splitting using density functional theory[22–25]. Although temperature variation is very common in our daily life, up to now, there is rare report on the hydrogen generation or dye decomposition using pyroelectric 2D-BP.

In this work, we demonstrate the pyro-catalytic behavior of 2D-BP via hydrogen evolution and dye decomposition in a room-temperature thermal cycling between 15 °C and 65 °C. Our work shows the great potential for energy harvesting from the cold–hot temperature variation using pyroelectric 2D materials.

## Results

**Characterization of 2D-BP.** X-ray diffraction (XRD) pattern of 2D-BP is shown in Fig. 2a, whose strong diffraction peaks suggest that the 2D-BP is well crystallized (JCPDS Card no. 73–1358 for black phosphorene). The scanning electron microscopy (SEM) image (inset of Fig. 2a) shows that thin and transparent 2D-BP nanosheets were spread on Si substrate. Generally, the catalytic efficiency depends greatly on the catalyst's particle size[26]. Small size and high surface area of nano-particles can result in easy migration of the pyroelectric charges between the pyroelectric materials and reactants[27], and hence the small particle size of 2D-BP ensures its excellent pyro-catalytic performance. Energy dispersive X-ray spectroscopy (EDS) in Fig. 2b shows the existence of phosphorous (81.33%), Al (3.96%), and Au (14.71%), where Al and Au come from the sample holder and the sputtered Au coating on the powder sample, respectively. Figure 2c gives the X-ray photoelectron spectroscopy (XPS) spectra of 2D-BP in a wide energy range, where all the binding energies are calibrated

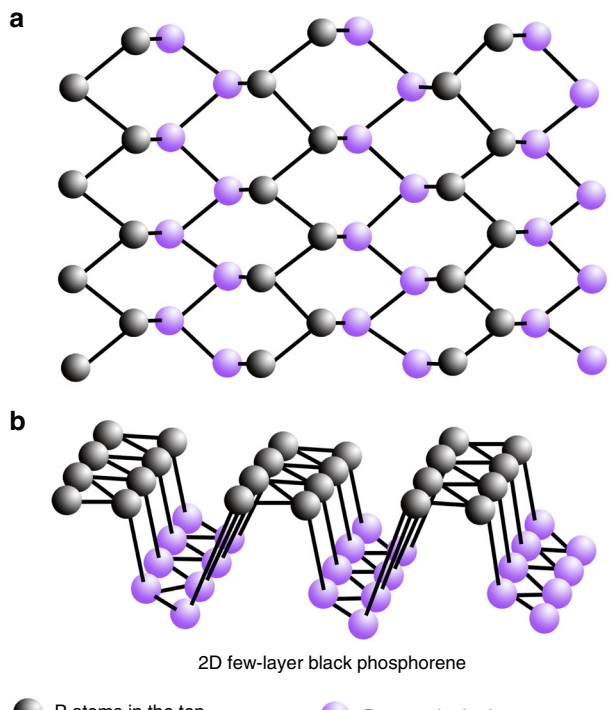

**a**

**b**

2D few-layer black phosphorene

● P atoms in the top    ● P atoms in the bottom

**Fig. 1** Schematic illustration of atomic structure of 2D-BP. **a** Top view. **b** 3D view

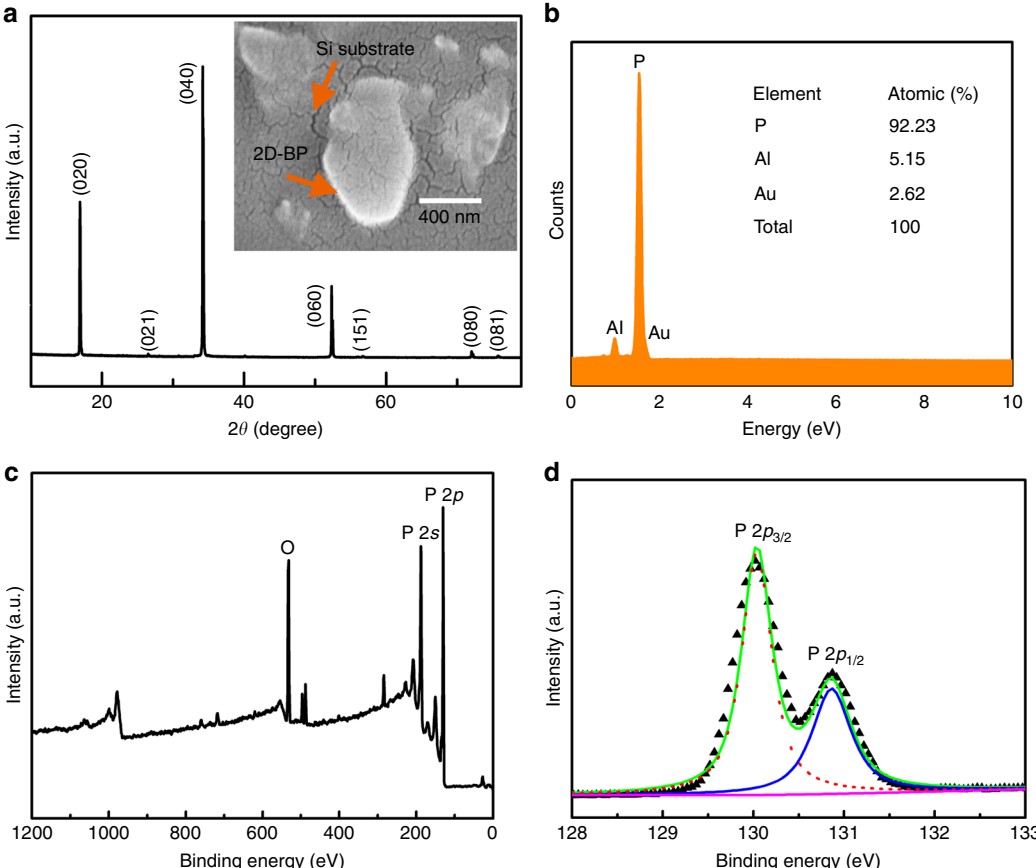

**Fig. 2** Microscopy and spectroscopy of 2D-BP. **a** XRD patterns. The inset is the SEM image. **b** EDS spectrum. **c** XPS spectrum in a wide energy range. **d** P 2p core level XPS spectrum

using the binding energy of C 1s. The oxygen contamination is due to the long time exposure to air[28]. No characteristic peaks of other contaminates, apart from oxygen, are found in the XPS spectrum. Figure 2d shows the P 2p core level spectrum of the 2D-BP, where the characteristic spin-orbital doublet can be observed with $2p_{3/2}$ at 130.02 eV and $2p_{1/2}$ at 130.87 eV[29].

Theoretically, 2D-BP with a non-centro-symmetric structure (m point group) is piezoelectric, which can be verified by the piezoresponse force microscopy (PFM). The topologies, vertical piezoresponse amplitude, and phase images of the 2D-BP nanosheets are shown in Fig. 3a–d, respectively. The topology images show clearly the nanosheet morphology of the 2D-BP. The amplitude and phase images also show clear contrasts. A hysteresis loop is recorded locally (Fig. 3d) and a 180° phase change occurs under the reversal of 15 V applied field, confirming a good piezoelectricity of the 2D-BP nanosheet. The dual alternating current resonance tracking (DART) modes were used in the PFM scanning in order to expel the electrostatic interaction contribution in displacement and the topographical interference in local electromechanical property mapping.

**Pyro-catalytic hydrogen evolution from water**. To study the generation of pyroelectric charges, the pyro-current response of 2D-BP under temperature fluctuation was measured. Figure 4a shows a typical temperature curve during the heating process, which was controlled by on/off of an infrared lamp. As shown in Fig. 4b, the pyro-current response agrees well with the slope of temperature change, which is generally described as[30],

$$I = p \cdot A \cdot (dT/dt), \quad (3)$$

where $I$, $A$ and $dT/dt$ are the pyro-current, the electrode coating area (9 cm$^2$), and the rate of temperature change, respectively. According to Eq. (3), the calculated pyroelectric coefficient $p$ is 5.287 mC m$^{-2}$ K$^{-1}$. For materials with a low thermal conductivity, fast temperature change ($dT/dt$) may result in a significant temperature gradient, which may lead to an additional thermoelectric signal, making it difficult to correctly evaluate the real contribution of pyroelectric effect in a pyro-catalytic experiment. Therefore, to minimize the disturbance of the thermoelectric effect, the pyroelectric coefficient ($p$) is often measured with a low $dT/dt$ rate of 0.045 °C s$^{-1}$ [31–33]. In the pyro-current measurement (Fig. 4) for the determination of pyroelectric coefficient, we adopted a similarly low $dT/dt$ rate of about 0.040 °C s$^{-1}$, while the thermoelectric effect could be neglected.

To be a potential candidate for pyro-catalytic hydrogen production, the catalyst should have a suitable electronic band structure. It has been reported that, for few-layer 2D-BP, the conduction band minimum is more negative than the $H^+/H_2$ reduction potential, according to density functional theory calculations[34], demonstrating the feasibility of 2D-BP for hydrogen production from water[35].

Furthermore, the pyro-potential $U$ built in a pyro-catalytic particle can be expressed as,

$$U = \frac{p \cdot \Delta T \cdot l}{\varepsilon}, \quad (4)$$

where $l$, $\Delta T$ and $\varepsilon$ are the size, the temperature change and the permittivity of a pyroelectric particle, respectively. $l$, $\Delta T$ and $\varepsilon$ of the monolayer 2D-BP are 0.35 nm, 50 °C, and 25 pC$^2$ N$^{-1}$ m$^{-2}$, respectively[36]. Using Eq. (4) and with the aid of the finite element

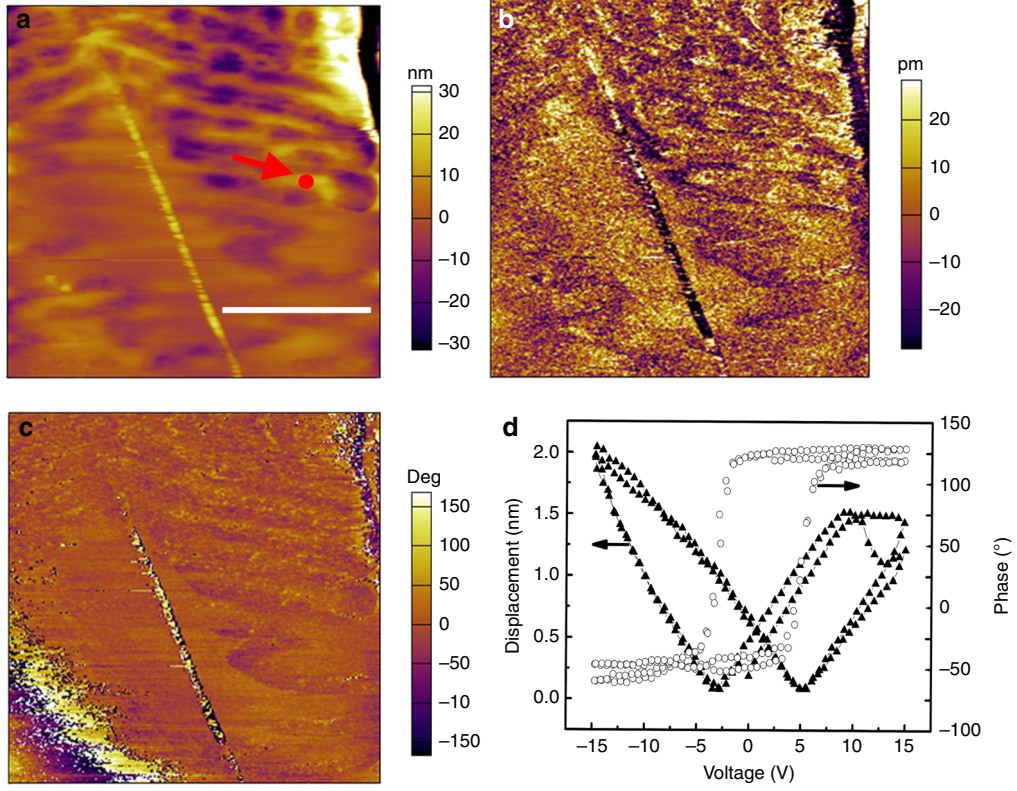

**Fig. 3** PFM of 2D-BP. **a** The morphology image. The scale bar is 2 μm. **b** The amplitude image. **c** The phase image. **d** Hysteresis loop

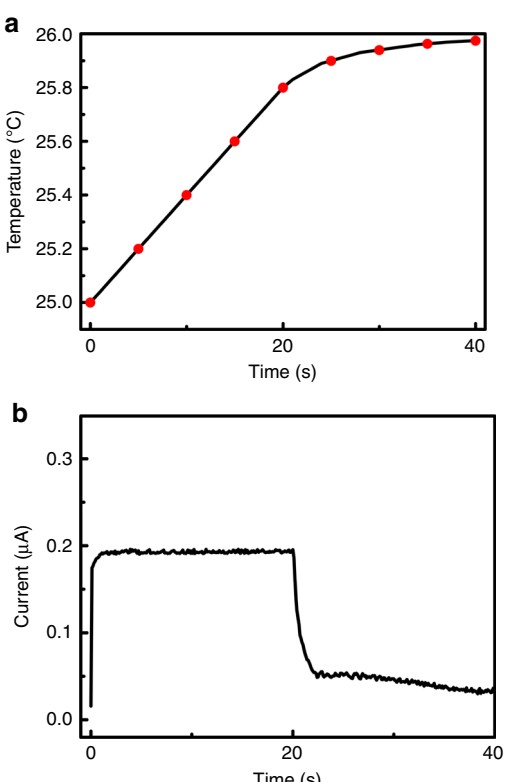

**Fig. 4** Pyro-current measurement. **a** The temperature curve in a heating process. **b** The corresponding pyro-current response

COMSOL software, the pyro-potential distribution across a 2D-BP nanosheet is shown in Fig. 5a, in which the pyro-potential changes from 0 to 3.7 V under the room-temperature 15–65 °C cold–hot alternation excitation.

The schematic diagram of pyro-catalytic hydrogen production is shown in Fig. 5b. The cold–hot alternation excitation can induce a net change of the electric dipole moment of the pyro-catalyst[37], which induces charge compensation on the pyro-catalyst surface. The pyro-generated positive charges ($q^+$) and negative charges ($q^-$) will transfer from the surface of pyro-catalyst to the reactant molecules to participate the redox reaction. The $H^+$ in water can react with the $q^-$ to form hydrogen, as shown in the following Eqs. (5) and (6)[38],

$$2D - BP \xrightarrow{\Delta T} 2D - BP(q^+ + q^-). \qquad (5)$$

$$2H^+ + 2q^- \rightarrow H_2. \qquad (6)$$

In theory, the rapid recombination of positive and negative carriers can seriously hinder the hydrogen generation efficiency[39,40]. In our experimental design, the sacrificial agent methanol reacts with positive charges and the hydroxyl ions, producing water molecules and hydroxyalkyl radical intermediate ($\cdot CH_2OH$)[41]. In a catalytic process, the addition of methanol sacrificial agent effectively scavenges the positive charge carriers, thereby increasing the lifetime of pyro-generated electrons and inhibiting the pyro-corrosion.

Sodium sulfide ($Na_2S$) and sodium sulfite ($Na_2SO_3$) can also be used as sacrificial agent[42]. The pyro-catalytic hydrogen evolution of the 2D-BP is around 17.8 μmol g$^{-1}$ per 15–65 °C thermal cycle with the addition of $Na_2S$ and $Na_2SO_3$ as sacrificial agent (Supplementary Fig. 1). No obvious hydrogen evolution occurs without the addition of sacrificial agent (Supplementary Fig. 1).

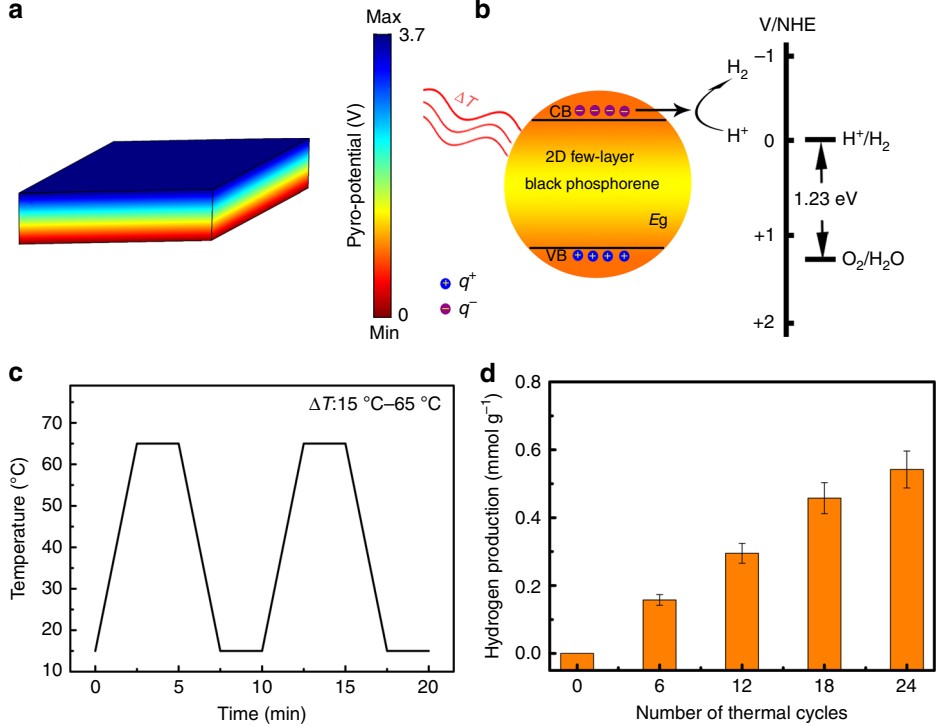

**Fig. 5** Pyro-catalysis of 2D-BP for hydrogen production from water splitting. **a** The COMSOL simulation of the pyro-potential. **b** Schematic diagram for hydrogen production through pyroelectric effect. **c** The temperature curve of the typical cold–hot thermal cycles from 15 to 65 °C. **d** Pyro-catalytic hydrogen production

Figure 5c shows the temperature curve of typical thermal cycles between 15 and 65 °C. The corresponding hydrogen production during cycling is shown in Fig. 5d, where the rate of hydrogen production is around 22.5 µmol g$^{-1}$ per thermal cycle. After 24 thermal cycles, the total hydrogen production per gram of catalyst is up to 0.54 mmol.

**Pyro-catalytic dye decomposition**. Besides the room-temperature pyro-catalytic hydrogen production, the 2D-BP was also tested for the pyro-catalytic decomposition of Rhodamine B (RhB) dye solution (5 mg L$^{-1}$), as shown in Fig. 6a. The RhB dye has the maximum absorption at 554 nm, which gradually decreases in magnitude with increasing number of thermal cycles. After 5 thermal cycles, RhB dye was almost completely decomposed.

The mechanism of pyro-catalytic dye decomposition is described as following: Under cold–hot temperature alternation excitation, the reactive oxygen species, such as superoxide anions (O$_2^{\cdot-}$) and hydroxyl radicals (·OH) are created at the surface of the pyro-catalyst through charge transfer of the pyroelectric charges. The pyro-catalytic dye decomposition reactions can be expressed in Eqs. 7–10,

$$2D-BP \xrightarrow{\Delta T} 2D-BP\,(q^+ + q^-), \tag{7}$$

$$O_2 + q^- \rightarrow O_2^{\cdot-}, \tag{8}$$

$$OH^- + q^+ \rightarrow \cdot OH, \tag{9}$$

$$\cdot OH\,(or\,O_2^{\cdot-}) + dye \rightarrow Decomposition. \tag{10}$$

We define pyro-catalytic decomposition ratio as $D = 1 - A_t/A_0$ ($A_t$ is the absorbance at time $t$ of the RhB solutions at $\lambda_{max} = 554$

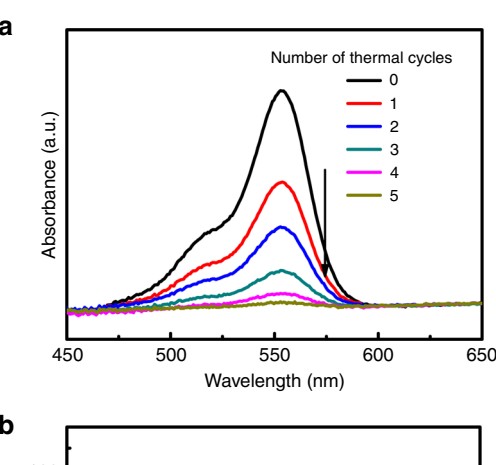

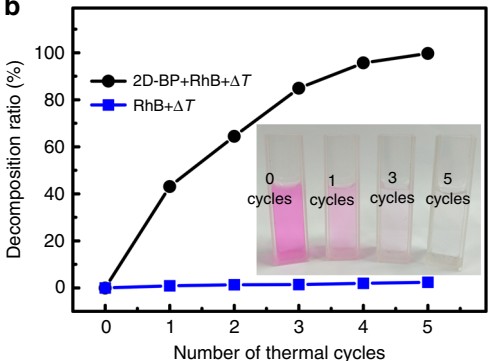

**Fig. 6** Pyro-catalysis of 2D-BP for dye decomposition. **a** The pyro-catalytic variation of the absorption spectra of RhB dye solution (5 mg L$^{-1}$) with different thermal cycles between 15 and 65 °C. **b** The decomposition ratio as a function of thermal cycles. The inset shows the photos of the pyro-catalytically-decomposed RhB dye solution of 2D few-layer black phosphorene (2D-BP) at different stages

nm and $A_0$ is the initial absorbance). It is found that $D$ quickly goes up to 99% after 5 thermal cycles (Fig. 6a). However, without the addition of the 2D-BP pyro-catalyst, the $D$ of RhB dye remains almost unchanged after 5 thermal cycles (Fig. 6b). Therefore, the simultaneous existence of the cold–hot alternation excitation and the 2D-BP catalyst is essential for pyro-catalysis, which shows the pyro-catalytic effect is originated from the combined pyroelectric effect and the electrochemical redox reaction, not from the direct thermal agitation on the dye solution. The pyro-catalytic effect of the 2D-BP can also be vividly viewed from the rapid color fading of RhB dye after a few thermal cycles (inset of Fig. 6b).

## Discussion

In general, the pyro-current shows a good linearity with rate of temperature change ($dT/dt$). It's reported that the pyro-current of 150-nm-think $PbZr_{0.2}Ti_{0.8}O_3$ film is proportional to $dT/dt$ over a large range from $10^{-2}$ to $10^3 \, °C \, s^{-1}$ rate[43]. The influence of the rate of temperature change on the pyro-catalytic dye decomposition of the 2D-BP was also investigated (Supplementary Fig. 2). The decomposition of RhB was complete after 9 thermal cycles with a $dT/dt$ of 0.17 °C s$^{-1}$ or 4 thermal cycles with a $dT/dt$ of 2.17 °C s$^{-1}$ (Supplementary Fig. 2). We have also observed the obvious pyro-catalytic RhB dye decomposition and hydrogen production from water in a decreasing temperature stage (Supplementary Fig. 3).

An additional pyro-catalytic experiment was also done through the use of traditional pyroelectric materials of $Pb(Mg_{1/3}Nb_{2/3})_{0.72}Ti_{0.28}O_3$ micro-sized crystals (Supplementary Fig. 4). $Pb(Mg_{1/3}Nb_{2/3})_{0.72}Ti_{0.28}O_3$ is one of the most prevalent pyroelectric materials with a high pyroelectric coefficient of 3.0 mC m$^{-2}$ K$^{-1}$ near room-temperature[44]. The $Pb(Mg_{1/3}Nb_{2/3})_{0.72}Ti_{0.28}O_3$ crystal grown in-house using a modified Bridgman technique and polarized along the <111> polarization axis was provided by Shanghai Institute of Ceramics of Chinese Academy of Sciences[44]. The $Pb(Mg_{1/3}Nb_{2/3})_{0.72}Ti_{0.28}O_3$ micro-crystals were prepared via milling the large crystals with a corundum mortar and pestle. The SEM (Supplementary Fig. 4a) and the XRD (Supplementary Fig. 4b) of $Pb(Mg_{1/3}Nb_{2/3})_{0.72}Ti_{0.28}O_3$ microcrystals were characterized, respectively. The pyro-catalytic RhB dye (5 mg L$^{-1}$) decomposition of $Pb(Mg_{1/3}Nb_{2/3})_{0.72}Ti_{0.28}O_3$ was obviously observed (Supplementary Fig. 4c). After experiencing 42 thermal cycles from 27 °C to 38 °C with the 0.036 °C s$^{-1}$ rate of temperature change, RhB dye is obviously decomposed (Supplementary Fig. 4c). Using fluorescent (FL) ·OH trapping agent terephthalic acid[45], we have detected the reactive oxygen species of ·OH in the pyro-catalytic dye decomposition process (Supplementary Fig. 4d). The amount of ·OH radicals almost increases linearly with thermal cycle numbers (Supplementary Fig. 4d), suggesting the stability of the ·OH production in pyro-catalytic dye decomposition of pyroelectric $Pb(Mg_{1/3}Nb_{2/3})_{0.72}Ti_{0.28}O_3$ materials.

Furthermore, the simultaneous excitation of heat and light has been reported as an effective way to improve catalytic efficiency[46]. As 2D-BP is widely reported to show excellent photocatalytic activity in hydrogen evolution[47–52], the current design for pyro-catalytic hydrogen generation can be further improved in the future by incorporating the photocatalytic function of 2D-BP, where the synergy between heat and light is expected to greatly enhance the overall hydrogen production.

In general, layer-by-layer etching will occur in BP after long-term exposure to air[53]. If there is no contact with oxygen, BP can be stable in water for a long period (several months)[54–56]. Actually, the application of BP as a photocatalyst or coating material for hydrogen generation from water has also been theoretically predicted and experimentally realized[57]. In our experiment, the evolution of oxygen was suppressed due to the use of sacrificial agents and hence the BP nanosheets were relatively stable in our pyro-catalytic reaction. The good linearity between the amount of hydrogen evolution and the number of thermal cycles implies that both the BP and its pyro-catalytic activity are stable during the whole experiment period. However, for large scale and long-term application where oxygen gas evolution is unavoidable, surface coating method should be utilized to stabilize the BP nanosheets[58].

In summary, under thermal cycling between 15 °C and 65 °C, the 2D few-layer black phosphorene shows a direct hydrogen generation of about 540 μmol per gram of catalyst after 24 thermal cycles and about 99% decomposition of RhB dye after 5 thermal cycles. The demonstrated pyro-catalytic effect of 2D-BP shows great potential in harvesting room-temperature cold–hot alternation heat energy for hydrogen generation from water and for dye decomposition.

## Methods

**Material preparation.** In general, most of the reported graphene-like 2D few-layer materials refers to materials whose thickness varies from a single layer to several layers[1,3]. In our work, the few-layer (~0–10 layers) 2D-BP nanosheets dispersed in deionized water (0.2 mg mL$^{-1}$) were commercially obtained (Nanjing XFNANO Mater. Tech. Co. Ltd., China). They were synthesized by the liquid-phase exfoliation method.

**Characterization.** X-ray diffraction was performed by a Philips PW3040/60 X-ray powder diffractometer equipped with a Cu Kα radiation ($\lambda = 1.54178$ Å) (the Netherlands). The morphology of 2D-BP was investigated by a Hitachi S-4800 scanning electron microscopy (Japan). Energy dispersive X-ray spectroscopy mapping was conducted for elemental analysis, using a Phenom ProX-EDS detector unit (the Netherlands). Chemical states of 2D-BP catalyst were determined by an ESCLALAB 250Xi X-ray photoelectron spectroscopy (USA).

The piezoelectric performance of the 2D-BP was measured by piezoresponse force microscopy (PFM, MFP-3D, USA), with a gold-coating cantilever (natural frequency of 75 kHz and force constant of 3 N m$^{-1}$). An alternate voltage of 2 V was applied between the PFM probe and the 2D-BP under different DC bias fields. The topology, vertical PFM amplitude and phase of the 2D-BP nanosheets were determined from the recorded nanosheet deformation response with the help of a laser interferometer and a lock-in amplifier.

**Hydrogen production experiments.** The pyro-catalytic hydrogen production of the 2D-BP was evaluated offline. In a typical experiment, 1 mg of the 2D-BP was dispersed in 10 mL of deionized water/methanol mixture (20 vol % methanol). Methanol was used as a sacrificial agent. The aqueous suspension sealed in a 25 mL borosilicate tube was evacuated and purged by Ar for about 5 min to completely remove air. The borosilicate tube was then transferred between heating water bath and cooling water bath. To detect the amount of hydrogen production, 1 mL gas component within the borosilicate tube was intermittently extracted and injected into a gas chromatograph (7890B, USA) with a thermal conductivity detector. The amount of hydrogen gas produced was calculated using a calibration curve of moles of hydrogen versus peak area.

**Pyro-electrochemical measurements.** The time-dependent pyro-current was recorded on an Chi660e-type electrochemical workstation (China) connected with a standard three-electrode system (Ag/AgCl reference electrode and Pt counter-electrode) placed in the electrolytic cell. The optically transparent SiO$_2$ conductive glass sheet was selected as the working electrode. The electrolyte solution was 50 mL Na$_2$SO$_4$ solution (0.5 M). Quantity of 0.5 mg 2D-BP was mixed with 10 μL ethanol and 10 μL Nafion perfluorinated resin solution and then spread on SiO$_2$ conductive glass sheet in about 3 × 3 cm$^2$. Then, the working electrode was dipped into the solution. Finally, the heating temperature excitation of the solution was controlled through turning on/off an infrared lamp fixed on the front side of the working electrode.

**Dye decomposition experiments.** Dye decomposition experiment was performed in a glass beaker. Before the measurement, the solution was completely stirred for 2 h to reach an adsorption-desorption equilibrium between dye and catalyst. Quantity of 1 mg 2D-BP was dispersed in 50 mL Rhodamine B dye solution (5 mg L$^{-1}$ in concentration) in a glass beaker, which was then put at a water bath center with continuous stirring to undergo thermal cycles in the dark. After each thermal cycle, 3 mL dye solution was centrifugally obtained. The concentration of dye solution in the pyro-catalytic decomposition process was monitored through

measuring the 554 nm absorption peak using a UV-vis spectrophotometer (Hitachi U-3900, Japan) and a calibration curve.

## Data availability

The data that support the findings of this study are available from the corresponding author upon request.

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

## Acknowledgements

This work was supported by the National Natural Science Foundation of China (51502266, 11574214, 11574126), the Public Welfare Technology Application Research Project of Zhejiang Province, China (LGG18E020005) and The Hong Kong Polytechnic University (1-ZVGH).

## Author contributions

Y.J. and Z.W. conceived and designed the research. H.Y. performed the material characterization. F.W. performed the XRD and PMF characterization. Z.W. and Y.W. characterized the pyro-catalysis of traditional pyroelectric material. Y.J. and H.H. performed the theoretical analysis. H.Y., Y.J. and H.H. wrote the paper with the help from all of the other co-authors.

## Additional information

**Competing interests:** The authors declare no competing interests.

