## [Peer Review File · Nature Communications]

Reviewers' comments:

Reviewer #1 (Remarks to the Author):

Interesting paper on the generation of hydrogen using the charge generated by the pyroelectric effect during temperature fluctuations of a 2D material. There is a need to improve the English throughout.

Technical comments:

Reference should be made to the work Kakekhani et al which relate DFT modelling of pyroelectric effects and electro-chemistry and water splitting (purely from modelling level - see papers below [1,2]) but related to a change in polarisation with temperature:

[1] A. Kakekhani and S. Ismail-Beigi, *Phys. Chem. Chem. Phys.*, 2016, 18, 19676–19695.

[2] A. Kakekhani and S. Ismail-Beigi, *J. Mater. Chem. A*, 2016, 4, 5235–5246

Ref [1] indicates the transition above and below the T_c is important and there is also been a recent review *Chem. Soc. Rev.*, 2017, 46, 7757-7786 which is worth adding to the introduction as it includes an overview of the limited work to date on pyro water splitting.

Hydrogen generation via pyroelectric effect on BaTiO₃ has also been recently published by R. Belitz, P. Meisner, M. Coeler, U. Wunderwald, J. Friedrich, J. Zosel, M. Schelter, S. Jachalke and E. Mehner, *Energy Harv. Syst.*, 2017, DOI: 10.1515/ehs-2016-0009.

Details of the approach to gas sampling should be more clear – for example, whether the gas tested sealed and allowed to gradually build up in concentration with time. The detection of both hydrogen and oxygen would also provide clarity of water splitting. It would also be good to have a control sample showing no H₂ build up.

Eqn 4 is fine and I have undertaken the calculation using the data of thickness nm, p measures, a ΔT of 18K and the permittivity given (the units used for permittivity on page 8) to get a 3.7V. This is fine, but I see no reason why a COMSOL model is needed to also show this as the information can simply be used in a simple equation?

There is a statement in voltage is greater than the 0.27V overvoltage, but in reality you need 1.23V plus the over-voltage - i.e. $1.23+0.27= 1.5V$ so this needs addressing.

Overall interesting work.

Reviewer #2 (Remarks to the Author):

In this manuscript, the authors utilized 2D layered black phosphorene as pyro-catalyst to generate hydrogen and degrade RhB dye under a cold-hot alternation temperature change. 8.4 $\mu\text{mol/g}$ hydrogen was generated after 36 cold-hot (22°C-40°C) cycles and nearly 97% RhB dye degradation was observed after 12 cycles. They also used PFM to characterize the fundamental piezoelectric (pyroelectric) property of 2D layered black phosphorene. In general, this concept proposed in this manuscript is interesting. But from the data the authors provided in this manuscript, it is hard to convince the readers that those effects are coming from pyroelectric property of 2D layered black phosphorene. For example, the authors purely used a water bath to create a temperature change and

the time they need to heat the solution from 22°C to 40°C was about 7.5 minutes. The slow temperature change process would not create a rate of change in temperature (dT/dt). In addition, from the output curve shown in Figure 3, it seems that this is thermoelectric output instead of a pyroelectric output. Therefore, I would not suggest to accept this manuscript unless they can further prove that these effects are really from pyroelectric property of 2D layered black phosphorene. And it will be better that they can also study typical pyroelectric materials to demonstrate that the pyro-catalytic effect indeed results in the hydrogen production and dye degradation, not possibly from the reactions with water/methanol and RhB dye. Also, if this is really from the pyroelectric property of 2D layered black phosphorene, then decreasing solution temperature would also result in the hydrogen generation and dye degradation. Therefore, they need to add the data by changing the solution temperature from ambient to lower temperature.

Reviewer #3 (Remarks to the Author):

From comprehensive experimental measurements, the authors have shown the interesting pyro-electric behavior and the possibility to convert thermal energy into electricity of black phosphorene under cold-hot alternation, which is then proposed to split water and decompose dye molecules. Therefore the 2D black phosphorenes are potentially used in energy harvesting. It is an interesting and import findings, but some points need to be clarified before its publication in nature communications

1. At the abstract part, the first sentence "recently, graphene-like superconductivity" is weird, I can't understand the meaning and purpose the authors mentioned piezoelectricity and superconductivity here, as the topic is not about these two issues, and also not related with MX2.
2. in the following sentence, it says graphene-like 2D layered black phosphorene, but actually BP is not graphene like, one can see it from figure 1, BP has buckling structures.
3. Equation 1, P_i at right side should be P_j ?
4. At line 78, it says the nanosheets have thickness of nm, but how many nanometers? it is important to show the readers the thickness, and estimate the layer number. And furthermore, it is necessary to show the thickness effect, namely how is the catalytic efficiency about water splitting and dye decomposition as a function of thickness.
5. at line 112-114, it says "confirming a good piezoelectricity, therefore good pyro-electric property"? What is the underlying relationship between these two feature, I didn't see the direct relationship between them.
6. From equations 6-8 at line 180-186, there is a hydroxyalkyl radical intermediate (CH_2OH), I am wondering where is the element C coming from? how is the intermediate CH_2OH produced?

Revision Notes

Reviewers' comments:

Reviewer #1 (Remarks to the Author):

1. Interesting paper on the generation of hydrogen using the charge generated by the pyroelectric effect during temperature fluctuations of a 2D material. There is a need to improve the English throughout.

We have carefully and thoroughly rewritten the whole manuscript to improve the English.

Technical comments:

1. Reference should be made to the work Kakekhani et al which relate DFT modelling of pyroelectric effects and electro-chemistry and water splitting (purely from modelling level - see papers below [1,2]) but related to a change in polarisation with temperature:

[1] A. Kakekhani and S. Ismail-Beigi, *Phys. Chem. Chem. Phys.*, 2016, 18, 19676–19695.

[2] A. Kakekhani and S. Ismail-Beigi, *J. Mater. Chem. A*, 2016, 4, 5235–5246

Ref [1] indicates the transition above and below the T_C is important and there is also been a recent review *Chem. Soc. Rev.*, 2017, 46, 7757-7786 which is worth adding to the introduction as it includes an overview of the limited work to date on pyro water splitting. Hydrogen generation via pyroelectric effect on BaTiO_3 has also been recently published by R. Belitz, P. Meisner, M. Coeler, U. Wunderwald, J. Friedrich, J. Zosel, M. Schelter, S. Jachalke and E. Mehner, *Energy Harv. Syst.*, 2017, DOI: 10.1515/ehs-2016-0009.

Response: Thanks for the Reviewer's helpful comments. Curie temperature T_C is a temperature at which ferroelectric materials lose their spontaneous polarizations to become paraelectric ones [*Science* 2008, **321**, 821; *J. Mater. Chem. A* 2016, **4**, 5235]. As mentioned by the reviewer, the ferroelectric-paraelectric phase transition could bring about strong pyroelectric effect near the Curie temperature T_C [*J. Mater. Chem. A* 2016, **4**, 5235; *Appl. Phys. Lett.* 2013, **103**, 182906; *Phys. Chem. Chem. Phys.* 2016, **18**, 19676], at which large pyroelectric current releases occur due to the quick decrease of polarization in a narrow temperature range [*Appl. Phys. Lett.* 2017, **110**, 102905; *Appl.*

Phys. Lett. 2012, **101**, 262901]. For 2D few-layer black phosphorene, no trace of phase transition can be identified from the temperature dependence of lattice parameters in the range of room temperature to 300 °C [J. Phys. Chem. Lett. 2015, **6**, 773], meaning that no phase transition occurs in our experimental temperature range of 15- 65 °C. It has also been reported that the black phosphorene monolayer is very stable and remains an ordered puckered hexagonal structure up to a temperature < 300 °C [Small 2015, **11**, 640].

Xie *et al.* have reported the coupling between pyroelectric effect and electro-chemical process and observed its water-splitting effect driven by these pyroelectric charges [*Int. J. Hydrogen Energ.* 2017, **42**, 23437]. Belitz *et al.* have investigated the BaTiO₃ microcrystalline powders for pyro-driven hydrogen generation [*Energy Harv. Syst.* 2017, **4**, 107]. Kakekhani *et al.* have theoretically studied the pyroelectrically-induced water splitting using density functional theory [*J. Mater. Chem. A* 2016, **4**, 5235; *ACS Catal.* 2015, **5**, 4537; *Surf. Sci.* 2016, **650**, 302; *Phys. Chem. Chem. Phys.* 2016, **18**, 19676]. Recently, 2D few-layer black phosphorene (2D-BP), has been reported as the magical materials behaving unexpected anisotropic optoelectronic and electronic properties, and has attracted global interest. Up to now, there is rare report on the hydrogen generation or dye decomposition using pyroelectric 2D-BP. This work highlights the potential application of 2D few-layer black phosphorene in harvesting room-temperature cold-hot alternation energy.

All the suggested references mentioned above have been cited in our revised manuscript.

2. Details of the approach to gas sampling should be more clear—for example, whether the gas tested sealed and allowed to gradually build up in concentration with time. The detection of both hydrogen and oxygen would also provide clarity of water splitting. It would also be good to have a control sample showing no H₂ build up.

Response: Based on the reviewer's suggestion, the following gas sampling process has been added in the Methods part of the revised manuscript.

The pyro-catalytic hydrogen production of the 2D-BP was evaluated offline. In a typical

experiment, 1 mg of the 2D-BP was dispersed in 10 mL of deionized water/methanol mixture (20 vol % methanol). Methanol was used as a sacrificial agent. The aqueous suspension sealed in a 25 mL borosilicate tube was evacuated and purged by Ar for about 5 min to completely remove air. The borosilicate tube was then transferred between heating water bath and cooling water bath. To detect the amount of hydrogen production, 1 mL gas component within the borosilicate tube was intermittently extracted and injected into a gas chromatograph (7890B, USA) with a thermal conductivity detector. The amount of hydrogen gas produced was calculated using a calibration curve of moles of hydrogen versus peak area.

We also tried the splitting of pure water (Fig.S1 in the “Supplementary Information”) and did observe the hydrogen generation. However, the hydrogen generation rate was very small. Sacrificial agents have been frequently used in photocatalysis reaction system to consume the photo-generated holes, thereby increasing the lifetime of photo-generated electrons and inhibiting the photo-corrosion to improve the efficiency [*Chem. Soc. Rev.* 2009, **38**, 253; *Angew. Chem., Int. Ed.*, 2013, **125**, 5746]. We adopted the same method in our pyro-catalysis study, where sacrificial agent methanol was added to consume the positive charges and hence reduce the charge recombination. Therefore, in our experiment, the main gas product is H₂, which is beneficial for the purpose of hydrogen fuel production.

A control sample without the addition of 2D-BP was also tested and the results are shown in Fig. 6b of the revised manuscript. It can be seen that, without the addition of 2D-BP, there is almost no decomposition of RhB after 5 thermal cycles.

The above discussion has been added in our revised manuscript.

3. Eq. 4 is fine and I have undertaken the calculation using the data of thickness nm, p measures, a delta T of 18K and the permittivity given (the units used for permittivity on page 8) to get a 3.7V. This is fine, but I see no reason why a COMSOL model is needed to also show this as the information can simply be used in a simple equation?

Response: Thanks for the reviewer's comments. The calculation process from Eq. 4 is very simple and the COMSOL result seems is not necessary. However, we still want to use the COMSOL simulation result in order to give an intuitive view of the pyro-potential distribution. The similar handling way has also been widely adopted in some references related to pyroelectric energy harvesting [*Nano Lett.*, 2012, **12**, 2833; *Nano Lett.*, 2012, **12**, 6408].

4. There is a statement in voltage is greater than the 0.27 V overvoltage, but in reality you need 1.23 V plus the over-voltage - i.e. $1.23+0.27= 1.5$ V so this needs addressing. Overall interesting work.

Response: Thanks for the reviewer's comments. We have corrected the corresponding sentence to 'which is much higher than the required potential > 1.5 V (1.23V plus the overvoltage 0.27 V)'.

To make this point clearer, we have added the following discussion in our revised manuscript.

To be a potential candidate for pyro-catalytic hydrogen production, the catalyst should have a suitable electronic band structure [*Chem. Soc. Rev.* 2015, **44**, 2060]. Density functional theory calculations [*J. Phys. Chem. C* 2014, **118**, 26560] showed that the bottom of the conduction band of few-layer 2D-BP is more negative than that of the reduction potential of H^+/H_2 , implying that 2D-BP is a good candidate for hydrogen production from water.

Reviewer #2 (Remarks to the Author):

In this manuscript, the authors utilized 2D layered black phosphorene as pyro-catalyst to generate hydrogen and degrade RhB dye under a cold-hot alternation temperature change. 8.4 $\mu\text{mol/g}$ hydrogen was generated after 36 cold-hot (22 $^{\circ}\text{C}$ - 40 $^{\circ}\text{C}$) cycles and nearly 97% RhB dye degradation was observed after 12 cycles. They also used PFM to characterize the fundamental piezoelectric (pyroelectric) property of 2D layered black phosphorene.

1. In general, this concept proposed in this manuscript is interesting. But from the data the authors provided in this manuscript, it is hard to convince the readers that those effects are coming from pyroelectric property of 2D layered black phosphorene. For example, the authors purely used a water bath to create a temperature change and the time they need to heat the solution from 22 $^{\circ}\text{C}$ to 40 $^{\circ}\text{C}$ was about 2.5 minutes. The slow temperature change process would not create a rate of change in temperature (dT/dt). In addition, from the output curve shown in Figure 3, it seems that this is thermoelectric output instead of a pyroelectric output. Therefore, I would not suggest to accept this manuscript unless they can further prove that these effects are really from pyroelectric property of 2D layered black phosphorene.

Response: Thanks for the reviewer's comments. Thermoelectricity and pyroelectricity are the two main methods to convert thermal energy to electrical energy. Thermoelectricity and pyroelectricity are two easily linked phenomena under the temperature fluctuation. However, their mechanisms are different. The thermoelectricity needs a temperature difference on both sides of the thermoelectric material (dT/dx), while the pyroelectricity only requires the temperature fluctuations time-dependent (dT/dt) [*Nano Lett.* 2012, **12**, 2833]. It is hard to design thermoelectricity of nanomaterials for pyro-catalysis because it is difficult to keep a temperature difference in nanoscale extent [*Nano Lett.* 2012, **12**, 2833].

It is true that a fast dT/dt may result in an significant temperature difference between the surface and inner of material due to the limit of thermal conductivity. In order to minimize the disturbance of the thermoelectric effect, the pyroelectric coefficient (p) is often measured with a low dT/dt rate of 0.045 K/s, as demonstrated by many groups [*Appl. Phys. Lett.* 2005, **86**, 082901; *Appl. Phys. Lett.* 2006, **89**, 162906; *J. Appl. Phys.* 2005, **98**, 084104; *J. Phys. D: Appl. Phys.* 2009, **42**, 075406; *J. Am. Ceram. Soc.* 2016, **99**, 1294]. Pyroelectric coefficient p is essentially unchanged for a wide dT/dt range from 0.018 $^{\circ}\text{C}/\text{s}$ to 10^3 $^{\circ}\text{C}/\text{s}$ [*J. Appl. Phys.* 2012, **112**, 104106; *Microelectron. Eng.* **1995**, 59, 1; *J. Appl.*

Phys. **2008**, 104, 114102; *J. Appl. Phys.* 2007, **101**, 054113]. We adopted a similarly low dT/dt (~ 0.040 °C/s) in the pyro-current measurement (Fig. 4 of the revised manuscript). Our pyro-catalysis is due to the pyroelectric effect and not due to the thermoelectric effect.

Fig.1 The absorption spectra of RhB dye solution (5 mg/L) after adding 2D few-layer black phosphorene experiencing different number of thermal cycles with dT/dt of (a) 0.17 °C/s; (b) 2.17 °C/s.

The rate of temperature change (dT/dt) is important to pyro-catalysis. The pyro-current can be described as $I = p \cdot A \cdot (dT/dt)$, where I , p , A and dT/dt are the pyro-current, pyroelectric coefficient, the electrode coating area of pyro-catalyst, and the rate of temperature fluctuation, respectively [*Nano Lett.* 2012, **12**, 2833]. It's reported that the pyro-current of 150-nm-thick $\text{PbZr}_{0.2}\text{Ti}_{0.8}\text{O}_3$ film is proportional to dT/dt over a large range from 10^{-2} to 10^3 °C/s rate [*J. Appl. Phys.* 2012, **112**, 104106]. On basis of the reviewer's suggestion, we have investigated the influence of rate of temperature change, as shown in Fig. 1 (Supplementary Fig. S2 in our revised manuscript). We

moved the sealed flat bag filled the mixture of 1 mg 2D-BP catalyst and 50 mL RhB dye solution with a concentration of 5 mg/L between the 65 °C hot and 15 °C cold water bath source. In this experiment, two rates of temperature change dT/dt are 0.17 °C/s and 2.17 °C/s, respectively. The faster dye decomposition was found under the higher dT/dt .

The total pyroelectric charge delivered during one thermal cycle can be calculated as $Q = p \cdot A \cdot \Delta T$, where p is the pyroelectric coefficient, A is the surface area of 2D-BP and ΔT is the temperature change. Therefore, if we use better exfoliated 2D-BP with lower average layer thickness (corresponding to a higher total surface area A per gram catalyst), more hydrogen can be generated per thermal cycle. Hence, we did the whole pyro-catalysis experiment again with better exfoliated 2D-BP (with more amount of 1~10 few-layers). To address the reviewer's concern, a slightly enhanced rate of temperature change of 20 °C/min (0.33 °C/s) was adopted for pyro-catalytic hydrogen evolution. As shown in Fig. 2, the hydrogen evolution increases to 25.4 $\mu\text{mol/g}$ per 15-65 °C thermal cycle, which is much higher than the rate in our original manuscript. We updated the whole experimental results in our revised manuscript. These improved results do not change the major conclusion of our manuscript. However, they are more attractive to researchers in this area.

Fig. 2 The hydrogen evolution with temperature change from 15-65 °C. The inset is the temperature curve of two thermal cycles.

2. And it will be better that they can also study typical pyroelectric materials to demonstrate that the pyro-catalytic effect indeed results in the hydrogen production and dye degradation, not possibly from the reactions with water/methanol and RhB dye.

Response: On basis of the suggestion of the reviewer, we studied the pyro-catalytic effect of some typical pyroelectric materials (Supplementary Fig. S4 in our revised manuscript). [Redacted]

[Redacted]

[Redacted]

[Redacted]

$\text{Pb}(\text{Mg}_{1/3}\text{Nb}_{2/3})_{0.72}\text{Ti}_{0.28}\text{O}_3$ is one of the most prevalent pyroelectric materials with a high pyroelectric coefficient of about $3.0 \text{ mC}/(\text{m}^2 \cdot \text{K})$ near room-temperature [*Mater. Sci. Eng. B* 2005, **119**, 71]. The $\text{Pb}(\text{Mg}_{1/3}\text{Nb}_{2/3})_{0.72}\text{Ti}_{0.28}\text{O}_3$ crystal grown in-house using a modified Bridgman technique and polarized along the $\langle 111 \rangle$ polarization axis was provided by Shanghai Institute of Ceramics of Chinese Academy of Sciences [*Mater. Sci. Eng. B* 2005, **119**, 71]. The $\text{Pb}(\text{Mg}_{1/3}\text{Nb}_{2/3})_{0.72}\text{Ti}_{0.28}\text{O}_3$ microcrystallines were prepared via milling the crystal with a corundum mortar and pestle.

Due to the broken inversion structure, 2D MoS_2 may also be ferro-/pyro-/piezoelectric. Many references have reported the piezoelectric performance of triangle few-layer 2D MoS_2 nanosheets or 2D MoS_2 nanoflowers [*Nature* 2014, **514**, 470; *Nature Nanotechnology* 2015, **10**, 151; *Nature Communications*, 2015, **6**, 7430; *Phys. Rev. B* 2011, **83**, 115328; *J. Phys. Chem. Lett.* 2012, **3**, 2871; *Nano Lett.* 2016, **16**, 849; *Adv. Mater.* 2016, **28**, 3718]. Here MoS_2 nanoflowers were synthesized via a hydrothermal method [*J. Power. Sources* 2015, **300**, 358]. In a typical preparation process, 3.7 g sodium molybdate ($\text{Na}_2\text{MoO}_4 \cdot 2\text{H}_2\text{O}$) and 0.675 g CH_3CSNH_2 (TAA) were dissolved into 30 ml deionized water, then 0.5 g sodium silicate ($\text{Na}_2\text{SiO}_3 \cdot 9\text{H}_2\text{O}$) was added into the solution under violent stirring. The pH value of the solution was adjusted to 6.0 by dropping 12 M HCl solution under violent stirring. The solution was transferred into a 50 ml Teflon-lined stainless steel autoclave and sealed tightly. Hydrothermal reaction was carried out at $220 \text{ }^\circ\text{C}$ for 24 h. After that, the autoclave was allowed to cool down naturally. The black precipitates were collected, washed first with 1 M NaOH solution for several times to remove possible residual silicic acid and then washed with deionized water and absolute ethanol, and finally dried at $60 \text{ }^\circ\text{C}$ for 6 h in a vacuum oven.

$\text{Ba}_{0.7}\text{Sr}_{0.3}\text{TiO}_3$ is a traditional pyroelectric material with a large pyroelectric coefficient of about $80 \times 10^{-8} \text{ C}/(\text{cm}^2 \cdot \text{K})$ [*Phys. Status Solidi A* 2011, **208**, 2699]. Here $\text{Ba}_{0.7}\text{Sr}_{0.3}\text{TiO}_3$ nanoparticles were prepared through a hydrothermal-assisted sol-gel method. Briefly, 1.788 g of $\text{Ba}(\text{CH}_3\text{COO})_2$ and 0.725 g of $\text{Sr}(\text{CH}_3\text{COO})_2$ with a molar ratio of 7:3 were dissolved in 20 mL of acetic acid at $80 \text{ }^\circ\text{C}$, respectively. The obtained colorless solutions were then mixed together with continuous stirring at a speed of 400 rpm for 15 min, marked as Solution A. Meanwhile, 3.52 mL of $\text{Ti}(\text{C}_4\text{H}_9\text{O})_4$ was added

into 16 mL of ethylene glycol monomethylether, obtaining yellowish-brown solution, which was added into Solution A dropwise. Then, 0.1 g of $C_{16}H_{33}(CH_3)_3NBr$ was added into the solution, which was then transferred to a Teflon cup with 80% capacity. Thereafter, the Teflon cup was sealed tightly in a stainless steel autoclave with continuous stirring at a speed of 500 rpm at 120 °C for 6 h until the solution changed from yellowish to milky white. After that, the solution was continuously stirred at 60 °C for 3 h in a beaker. The final sample was annealed at 500 °C for 1 h and 800 °C for 3 h and ball-milled into powder before characterization.

3. Also, if this is really from the pyroelectric property of 2D layered black phosphorene, then decreasing solution temperature would also result in the hydrogen generation and dye degradation. Therefore, they need to add the data by changing the solution temperature from ambient to lower temperature.

Response: As suggested by the reviewer, we also did the pyro-catalytic experiments only during decreasing temperature.

Fig. 5a shows the pyro-catalytic dye decomposition of the 2D-BP at different time t_1 , t_2 , t_3 and t_4 . In this experiment, we moved the sealed flat bag (filled the mixture of catalyst and dye solution) between the hot and cold water bathes. The temperature change curve is shown in the inset of Fig. 5a. Pyro-catalytic dye decomposition can be observed both in the heating-up and cooling-down stages.

We also measured pyro-catalytic hydrogen evolution of the 2D-BP during the cooling stages, as shown in Fig. 5b. The temperature change curve is shown in the inset of Fig. 5b. Obvious pyro-catalytic hydrogen evolution was observed during the cooling stage.

Fig. 5 (a) Pyro-catalytic dye decomposition of 2D-BP at different time t_1 - t_4 , which are shown in the heating-up and cooling-down temperature curve (inset). (b) Pyro-catalytic hydrogen evolution of 2D-BP. The inset is the cooling-down temperature curve.

These experimental results and corresponding discussion have been added in the “Supplementary Information” of our revised manuscript.

Reviewer #3 (Remarks to the Author):

From comprehensive experimental measurements, the authors have shown the interesting pyroelectric behavior and the possibility to convert thermal energy into electricity of black phosphorene under cold-hot alternation, which is then proposed to split water and decompose dye molecules. Therefore the 2D black phosphorenes are potentially used in energy harvesting. It is an interesting and import findings, but some points need to be clarified before its publication in nature communications

1. At the abstract part, the first sentence "recently, graphene-like superconductivity" is weird, I can't understand the meaning and purpose the authors mentioned piezoelectricity and superconductivity here, as the topic is not about these two issues, and also not related with MX_2 .

Response: The description of “Recently, graphene-like MX_2 (M= Mo, Nb, W; X= S, Se, Te)... superconductivity” is not directly related to the topic of this manuscript and is therefore deleted.

2. in the following sentence, it says graphene-like 2D layered black phosphorene, but actually BP is not graphene like, one can see it from figure 1, BP has buckling structures.

Response: Thanks for the reviewer’s comment. Just like graphene, phosphorus can form a 2D sheet. Although up-close, it is heavily corrugated while graphene is flat, being resulted from its different bonding structure with that of graphene. We hereby change the word “graphene-like” to a more precise one, “puckered graphene-like” [*Sci Rep.* 2014, **4**, 6946; *Small*, 2015, **11**, 640].

3. Equation 1, P_i at right side should be P_j ?

Response: Thanks. It is corrected in our revised manuscript.

4. At line 78, it says the nanosheets have thickness of nm, but how many nanometers? it is important to show the readers the thickness, and estimate the layer number. And furthermore, it is necessary to show the thickness effect, namely how is the catalytic efficiency about water splitting and dye

decomposition as a function of thickness.

Response: In general, most of the reported graphene-like 2D layered materials are referred as few-layer (one to several layers) materials [*Nature* 2005, **438**, 197; *Science* 2017, **357**, 306; *Nat. Nanotechnol.* 2014, **9**, 372].

In general, the band gap can obviously affect the catalytic performance. The band gap of the 2D few-layer black phosphorene can be tuned from 2.0 to 0.3 eV on basis of the first principles calculation [*Nanomaterials*, 2016, **6**, 194], depending on the number of layers [*Phys. Rev. B* 2014, **89**, 235319; *Phys. Rev. B* 2014, **89**, 201408]. For few-layer phosphorene, interlayer interactions reduce the band gap for each layer added, and eventually reach about 0.3 eV for bulk black phosphorus [*Nat. Nanotechnol.* 2014, **9**, 372; *Science* 2015, **349**, 723; *Adv. Funct. Mater.* 2015, **25**, 6996]. This thickness dependence of the band gap results from the quantum confinement of charge carriers in the out-of-plane direction [*Nat. Commun.* 2014, **5**, 4475].

In our work, the 2D few-layer black phosphorene nanosheets (0.2 mg/mL in deionized water) were synthesized by the liquid-phase exfoliation of bulk black phosphorous crystals and were bought from Nanjing XFNANO Mater. Tech. Co. Ltd., China with a thickness of 0.35-3.5 nm (about 0-10 layers). The scanning electron microscopy image (inset of Fig. 2a in our revised manuscript) shows that thin and transparent 2D-BP nanosheets were spread on Si substrate. In theory, the layers of 2D few-layer black phosphorene material can affect the pyro-catalysis. It should be noted that it is difficult to synthesize 2D few-layer black phosphorene with controlled number of layers via liquid exfoliation method [*Chem. Commun.* 2014, **50**, 13338]. Hence what we have a mixture of 2D-BP in different layers, which makes the thickness dependence study impossible at this stage. We also tried bulk black phosphorus, which did not show any pyro-catalytic dye decomposition.

We have added the discussion text in our revised manuscript.

5. at line 112-114, it says "confirming a good piezoelectricity, therefore good pyroelectric property"? What is the underlying relationship between these two feature, I didn't see the direct relationship

between them.

Response: Thanks for the reviewer's comments. Our previous statement is not precise. As shown in Fig. 6 below, all ferroelectrics are pyroelectrics, while all pyroelectrics are piezoelectrics [Energy Environ. Sci. 2014, 7, 25; J. Chem. Phys. 1994, 101, 5262]. The 2D-BP has a non-centrosymmetric structure with an *m* point group, which shows ferroelectricity. Hence the material we used is both piezoelectric and pyroelectric. The hysteresis loop (Fig. 3d in our revised manuscript) shown in the piezoresponse force microscopy (PFM) is just an indirect proof of pyroelectricity. The pyroelectricity was also confirmed by the pyro-current measurement (Fig. 4 in our revised manuscript). We have revised some of the expressions in our revised manuscript to make them more accurate.

Fig. 6 The relationships of Piezoelectric, pyroelectric and ferroelectric material [Energy Environ. Sci. 2014, 7, 25].

6. From equations 6-8 at line 180-186, there is a hydroxyalkyl radical intermediate (CH_2OH), I am wondering where is the element C coming from? how is the intermediate CH_2OH produced?/

Response: Enormous studies have demonstrated that photocatalyst suffers from rapid recombination of photogenerated carriers and severe photocorrosion, which seriously hinders the catalytic efficiency improvement [Chem. Soc. Rev. 2009, 38, 253; Angew. Chem., Int. Ed. 2013, 52, 5636]. To solve these problems, sacrificial agents have been frequently added into the reaction system to consume the photogenerated holes, thereby increasing the survival time of photogenerated electrons

and inhibiting the photocorrosion. The same method was adopted in the current pyro-catalytic reaction. To reduce the recombination of positive and negative charges, sacrificial agent methanol was added. In our pyro-catalytic water-splitting experiment, the main gas product is H₂. The sacrificial agent methanol reacts with these positive charges induced by the pyroelectric effect, producing hydrogen ion (H⁺) and a hydroxyalkyl radical intermediate ($\cdot\text{CH}_2\text{OH}$) [*Ind. Eng. Chem. Res.* 2013, **52**, 61].

Sodium sulfide (Na₂S) and sodium sulfite (Na₂SO₃) can also be used as sacrificial agent [*Catal. Today* 2013, **199**, 42]. The pyro-catalytic hydrogen evolution of the 2D-BP is around 17.8 μmol/g per 15-65 °C thermal cycle with the addition of Na₂S and Na₂SO₃ as sacrificial agent (supplementary Fig. S1 of our revised manuscript). No obvious hydrogen evolution occurs without the addition of sacrificial agent at cycle (supplementary Fig. S1).

In our original manuscript, we made a mistake to indicate the ~200 eV peak as C peak in the XPS curve. C element should have a binding energy of ~285 eV [*ChemElectroChem*, 2015, **2**, 324] in XPS curve. The peak located at the binding energy of ~200 eV in XPS curve belongs to P2s. We have corrected this mistake in our revised manuscript.

Reviewers' comments:

Reviewer #1 (Remarks to the Author):

I am happy with the revisions to clarify my own reviewer comments and those of the other two reviewers. The additional discussion and experimental results have improved the paper.

Chris Bowen

Reviewer #2 (Remarks to the Author):

I appreciate that the authors have made great efforts to improve the quality of this manuscript. However, before the acceptance of this manuscript, I suggest the authors can add one more data to clarify that the catalytic activity is really due to the pyroelectric effect. The authors just showed the catalytic results of raising the solution temperature, how about decreasing the solution temperature? If this catalytic activity is generated from the pyroelectric effect, the authors should observe same results when decreasing the solution temperature.

Reviewer #3 (Remarks to the Author):

From the revised manuscript, it is pleased to see that all the raised questions in the first round review have been well addressed with the satisfied evidences and supplementary experiments. Still one major problem should be addressed before its publication

It is well known that BP is unstable, and easy to be degraded under the presence of the moisture (water) and oxygen (there are too many investigations along the direction, I believe the authors are aware about this). In this work, hydrogen will be produced from water splitting, where the oxygen went? If combined with BP, it will be severely degraded, and the life time for hydrogen production will be quite short in this case.

Revision Notes

Reviewers' comments:

Reviewer #1 (Remarks to the Author):

I am happy with the revisions to clarify my own reviewer comments and those of the other two reviewers. The additional discussion and experimental results have improved the paper.

Response: Thanks a lot for your acceptance of our paper.

Reviewer #2 (Remarks to the Author):

I appreciate that the authors have made great efforts to improve the quality of this manuscript. However, before the acceptance of this manuscript, I suggest the authors can add one more data to clarify that the catalytic activity is really due to the pyroelectric effect. The authors just showed the catalytic results of raising the solution temperature, how about decreasing the solution temperature? If this catalytic activity is generated from the pyroelectric effect, the authors should observe same results when decreasing the solution temperature.

Response: Thanks for the reviewer's valuable comments. The result of pyro-catalytic experiment during decreasing temperature was shown in Fig.S3 of the Supplementary Information. For your easy reference, we copy-and-paste below.

Fig. 1a shows the pyro-catalytic dye decomposition of the 2D-BP at different time t_1 , t_2 , t_3 and t_4 . The temperature change curve is shown in the inset of Fig. 1a. Pyro-catalytic dye decomposition can be observed both in the heating-up (from t_1 to t_2) and cooling-down (from t_3 to t_4) stages. Particularly, we also measured the pyro-catalytic hydrogen evolution of the 2D-BP during the cooling stages *only*, as

shown in Fig. 1b. The temperature change curve is shown in the inset of Fig. 1b. Comparing the different time point t_5 and t_6 in the cooling stage, obvious pyro-catalytic hydrogen evolution was observed during the cooling stage only.

On the basis of the above experimental result, the pyroelectrically-induced pyro-catalysis occurs both in the heating and cooling stages, while no pyro-catalysis for the holding-temperature stage (from t_2 to t_3 in the inset of Fig. 1a, and from 0 to t_5 in the inset of Fig.1b) was observed.

Fig. 2 (a) Pyro-catalytic dye decomposition of 2D-BP at different time t_1 - t_4 , which are shown in the heating-up and cooling-down temperature curve (inset). (b) Pyro-catalytic hydrogen evolution of 2D-BP. The inset is the cooling-down temperature curve.

These additional experimental result and description for decreasing temperature stage have been added in the Supplementary Information of our 2nd revised manuscript.

Reviewer #3 (Remarks to the Author):

From the revised manuscript, it is pleased to see that all the raised questions in the first round review have been well addressed with the satisfied evidences and supplementary experiments. Still one major problem should be addressed before its publication. It is well known that BP is unstable, and easy to be degraded under the presence of the moisture (water) and oxygen (there are too many investigations along the direction, I believe the authors are aware about this). In this work, hydrogen will be produced from water splitting, where the oxygen went? If combined with BP, it will be severely degraded, and the life time for hydrogen production will be quite short in this case.

Response: Thanks for the reviewer's comments. In this study, we use sacrificial agents in the reaction system to consume the photogenerated holes, increase the lifetime of photogenerated electrons, and inhibit the photocorrosion. This is also a common way used in photocatalytic studies [*Chem. Soc. Rev.* 2009, **38**, 253; *Angew. Chem., Int. Ed.* 2013, **52**, 5636]. In our photocatalytic experiment, H^+ in water will first react with negative charges q^- to produce hydrogen. The hydroxyl ions from water can absorb positive charges q^+ to form hydroxyl radicals ($\cdot OH$), which further react with sacrificial agent CH_3OH to obtain hydroxyalkyl radical intermediate $\cdot CH_2OH$ and water, as shown in the Eq. (1)-(3) [*Ind. Eng. Chem. Res.* 2013, **52**, 61; *Nat. Commun.*, 2018, DOI: 10.1038/s41467-018-03543-y]. Thus, in our photocatalytic water-splitting experiment, there is almost no oxygen generated.

The role of oxygen and water in BP degradation has been intensively studied recently. It has been shown that the degradation of BP under ambient conditions is

initiated by contact with oxygen, while water does not play a primary role in the reaction [*Con-mat.mtrl-sci* arXiv: 1511.09201; *Chem. Mater.* 2016, **28**, 8330; *Nat. Commun.* 2015, **6**, 6647; *Angew. Chem.* 2016, **128**, 1]. After long-term exposure to air, there is a layer-by-layer etching of the BP [*2D Mater.* 2015, **2**, 011002]. Water is capable of removing P_xO_y from the surface and exposing P^0 for further oxidation. So, when there is simultaneous existence of oxygen and water, degradation and breakdown will happen (a process that takes several hours to days) [*2D Mater.* 2015, **2**, 011002]. If there is no oxygen, BP can be stable for a longer period. The BP nanosheets we bought are dispersed in water and can be stable for about 30 days, as supported by the document from the BP nanosheet supplier (Nanjing XFNANO Mater. Tech. Co. Ltd., China). In our experiment, according to the above equations (1-3), the oxygen is not the final catalytic reaction product and hence BP nanosheets can be maintained relatively stable in water.

Actually, the application of BP as a photocatalyst or coating material for hydrogen generation from water has been theoretically predicted and experimentally realized [*Energy Environ. Sci.*, 2016, **9**, 1513; *Angew.Chem.Int. Ed.* 2017, **129**, 2096; *Nat. Commun.* 2018,**9**, 1397], which means that BP is quite stable in the water during the hydrogen generation process.

Of course, we agree with the reviewer, there is always a risk due to BP's unstable behavior. Researchers are working towards the solutions to this problem [*Angew. Chem. Int. Ed.* 2016, **55**, 5003; *ACS Nano* 2015, **9**, 10411; *Phys. Rev. B* 2015, **91**, 085407]. For example, Zhao *et al* have developed a surface coordination strategy to enhance the stability of BP in air and water by preventing oxidization of BP [*Angew .Chem. Int. Ed.* 2016, **55**, 5003]. In the future, we hope to address the stability issue of BP. However, this is beyond the scope of the current study.

Reviewers' comments:

Reviewer #2 (Remarks to the Author):

It is pleased to see the response from the authors. Now I do not have major concerns, one minor concern is the stability of the BP the authors used in this paper. For example, is the weight of BP always maintaining the same during cycled tests? Or is the piezoresponse form BP remaining even after the treatment of 24 thermal cycles?

Reviewer #3 (Remarks to the Author):

All the questions from my side are already well addressed. Based on the interesting finding and strong theoretical and experimental supports to the conclusions, i would like to recommend its publication in Nature communications. The work is expected to inspire more research work on different materials along the direction.

Revision Notes

Reviewers' comments:

Reviewer #2 (Remarks to the Author):

It is pleased to see the response from the authors. Now I do not have major concerns, one minor concern is the stability of the BP the authors used in this paper. For example, is the weight of BP always maintaining the same during cycled tests? Or is the piezoresponse from BP remaining even after the treatment of 24 thermal cycles?

Response: Thanks for the reviewer's valuable comments. The amount of BP used in the photocatalytic cycling is very small and it is very difficult to accurately measure the weight of BP after drying. For the BP stability issue, we would like to address the following points.

(1) **Literature:** It has been shown that the degradation of BP under ambient conditions is initiated by contact with oxygen, while water does not play a primary role in the reaction [*Con-mat.mtrl-sci arXiv: 1511.09201; Chem. Mater.* 2016, **28**, 8330; *Nat. Commun.* 2015, 6,6647; *Angew. Chem.* 2016,**128**,1]. After long-term exposure to air, there is a layer-by-layer etching of the BP [*2D Mater.* 2015, **2**, 011002]. Water is capable of removing P_xO_y from the surface and exposing P^0 for further oxidation. So, when there is simultaneous existence of oxygen and water, degradation and breakdown will happen (a process that takes several hours to days) [*2D Mater.* 2015, **2**, 011002]. If there is no oxygen, BP can be stable for a longer period. Actually, the application of BP as a photocatalyst or coating material for hydrogen generation from water has been theoretically predicted and experimentally realized [*Energy Environ. Sci.*, 2016, **9**, 1513; *Angew.Chem.Int. Ed.* 2017, **129**, 2096; *Nat. Commun.* 2018,**9**, 1397].

(2) **Our sample:** The BP nanosheets we bought are dispersed in water and can be stable for about 30 days, as supported by the document from the BP nanosheet supplier (Nanjing XFNANO Mater. Tech. Co. Ltd., China). In our study, Ar was purged to completely remove air before catalytic experiment and sacrificial agents were used in the reaction system to consume the photogenerated holes. There is almost no oxygen existed or generated in our experiment and hence BP nanosheets can be maintained relatively stable in water.

(3) **Our experiment:** As shown in the figure below (Fig.1), there is a good linearity between the amount of hydrogen evolution and the number of thermal cycles, indicating that the amount of hydrogen generated per cycle is constant throughout the whole experiment period. This result also implies that the BP is stable and its pyrocatalytic activity is also stable during the whole experiment period.

Fig. 1 The hydrogen evolution as a function of number of thermal cycling between 15 and 65 °C. The straight line is a guide to the eye showing a linear relation between hydrogen evolution and number of thermal cycles.

(4) Of course, we agree with the reviewer that, for long term and large scale application, there is always a risk due to BP's unstable behavior after contacting oxygen. Researchers are working towards the solutions to this problem [*Angew. Chem. Int. Ed.* 2016, **55**, 5003; *ACS Nano* 2015, **9**, 10411; *Phys. Rev. B* 2015, **91**, 085407]. For example, Zhao *et al* have developed a surface coordination strategy to enhance the stability of BP in air and water by preventing oxidization of BP [*Angew. Chem. Int. Ed.* 2016, **55**, 5003]. Therefore, it is possible to enhance the stability of BP through these typical methods in future.

(5) Last, but not least, we hope this work can open avenues for pyrocatalytic hydrogen generation, where thousands of pyroelectric materials can be explored for more efficient and more stable hydrogen evolution.

We have added some discussion on the stability issue of BP in our revised manuscript.

Reviewer #3 (Remarks to the Author):

All the questions from my side are already well addressed. Based on the interesting finding and strong theoretical and experimental supports to the conclusions, i would like to recommend its publication in Nature communications. The work is expected to inspire more research work on different materials along the direction.

Response: Thanks a lot for your acceptance of our paper.

REVIEWERS' COMMENTS:

Reviewer #2 (Remarks to the Author):

The authors have well responded my questions and comments. Based on the novelty of this manuscript, I would like to recommend its publication in Nature Communications.